# The Role of Implicit Memory in the Development and Recovery from Trauma-Related Disorders

## Louis F. Damis

Integrative Health Psychology, PA, Oviedo, FL 32765, USA; DrDamis@LouisDamisPhD.com; Tel.: +1-407-697-8584

**Abstract:** Post-traumatic Stress Disorder is a chronic condition that occurs following a traumatic experience. Information processing models of PTSD focus on integrating situationally triggered sensory-emotional memories with consciously accessible autobiographical memories. Review of the nature of implicit memory supports the view that sensory-emotional memories are implicit in nature. Dissociation was also found to be associated with the development and severity of PTSD, as well as deficits in autobiographical memory. Moreover, disorganized attachment (DA) was associated with greater degrees of dissociation and PTSD, and like the defining neural activation in PTSD, was found to be associated with basal ganglia activity. In addition, subcortical neuroception of safety promotes a neurophysiological substrate supportive of social engagement and inhibition of fear-based responses. Furthermore, activation of representations of co-created imagined scenes of safety and secure attachment are associated with increases in this neurophysiological substrate. Repeated priming of secure attachment imagery was associated with modification of internal working models of DA along with reductions in dissociation and recovery from complex PTSD. In conclusion, it is posited that adequate recovery from extensive trauma experiences requires more than conscious elaboration of traumatic autobiographical memories and that the application of implicit nonconscious memory modification strategies will facilitate more optimal recovery.

**Keywords:** implicit memory; post-traumatic stress disorder; information processing; priming; disorganized attachment; internal working models; dissociation; polyvagal safety; phase-oriented treatment

## 1. Introduction

Posttraumatic stress disorder (PTSD) is an often chronic condition consequent to the experience of significant trauma [1]. One of the defining features of PTSD is the intrusive distressing recurring recollections of the traumatic event in the form of flashbacks, nightmares, and thoughts. Ehlers et al. (2004) noted that intrusive memories mainly consist of brief sensory fragments of the traumatic experience and often involve visual or somatic sensations [2]. These authors reported that intrusive reexperiencing symptoms often occur in the form of dissociative flashbacks whereby the individual loses awareness of the present surroundings and has the sense of reliving the traumatic event devoid of an appreciation of its original context. Additionally, contributing to intrusive reexperiencing are strong perceptual priming and associative learning of sensory information. Supporting an increase in associative learning, Miller et al. (2017) found an associative information processing bias in subjects with PTSD that interfered with hippocampal-dependent processing in active navigation [3]. Ehlers and Clark (2000) further relate the maintenance of PTSD to the failure of sensory information to become contextualized and integrated with normal continuous autobiographical memory, a process associated with reductions in distressful reexperiencing [4]. These authors noted that this failure to contextualize is partly related to disruptions in the encoding of episodic memory at times of extreme stress, rendering volitional recall of the traumatic experience fragmented, disorganized, poorly elaborated, not integrated into time and place, and incomplete. Elevated physiological arousal interferes with conceptual processing of episodic memory, impairing abstraction and meaning-making. This

cognitive information processing model of PTSD also underscores how memory processing deficits contribute to cognitive appraisals of trauma that contribute to the persistence of the disorder.

Considerable evidence supports the view that hippocampal function is disrupted in subjects with trauma-related disorders. The ability to mentally take multiple spatial perspectives of a scene, allocentric processing, is known to be dependent on hippocampal functioning. In contrast, an egocentric perspective, recalling a scene from the perspective in which it was experienced is not dependent on hippocampal functioning and is considered more driven by sensory-dependent memory processes. Smith et al. (2015) and Miller et al. (2017) found that allocentric spatial processing was impaired in subjects with PTSD supporting the disruptive functioning of hippocampal contextual processing [3,5]. Sierk et al. (2019) reported that script-driven traumatic imagery was of lesser severity for women with PTSD due to childhood trauma with higher allocentric spatial memory abilities [6]. Moreover, Bisby et al. (2010), in an analog study, demonstrated that low doses of alcohol while viewing a trauma video was associated with disruption in allocentric spatial processing, that egocentric processing remained intact, and that decrements in allocentric processing were associated with increases in traumatic intrusions over the next week [7]. Bisby et al. (2016) also found in a study of associative learning that the presentation of negative images increased memory for that item, but decreased associative learning [8]. These authors reported that hippocampal activity was reduced by the presence of negative images, but that amygdala activity increased for correctly retrieved negative images. Thus, negative emotional arousal disrupts hippocampal contextualization, associative learning in this case, and promotes amygdala fear learning.

Brewin et al. (2014) proposed a dual representation theory of PTSD that paralleled the cognitive model of Ehlers and Clark, noting that many types of psychopathology involve distressing involuntary images and visual memories [9,10]. These authors posit two distinct imagery and memory systems: the Sensory Representational System (S-Reps) and the Contextualized Representational System (C-Reps). The S-Reps system involves perceptual near-sensory memories, is automatically activated, and is relatively inflexible. In contrast, the C-Reps system is verbally accessible, voluntary, corresponds to the focus of conscious attention, and can adopt multiple viewpoints. C-Reps support contextualized episodic memories and verbal accounts. Moreover, "during traumatic events, encoding of S-Reps (perceptual memories) is strengthened whereas the encoding of C-Reps (contextualized episodic memories), and the connections between the S-Reps and C-Reps, is weakened" [10] (p. 88). Similar to the work of Ehlers et al., "contextualization refers to a process whereby selective attention leads to recording the sensory input into an abstract structural description. This recording then permits interaction with other knowledge, better organized consciously accessible memories, and reduced involuntary intrusions" (p. 88).

Brewin et al., noting the strong visual component to trauma-related intrusions, states that the flexible, consciously accessible context-dependent representations (C-Reps) project through the ventral visual stream to the inferior temporal cortex with its connections to the parahippocampus and hippocampus [9]. In contrast, these authors indicated that the inflexible, sensation-bound, viewpoint-dependent, involuntarily reactivated representation accompanied by strong emotional and autonomic components (S-Reps) project through the dorsal visual stream to superior parietal areas with their connections to the amygdala and insula. Moreover, they propose that these two types of representations interact via egocentric imagery in the precuneus and can be activated either top-down or bottom-up by either system, respectively.

## 2. Implicit and Explicit Memory

Bargh and Chartrand (1999) reported, "that most of the person's everyday life is determined not by their conscious intentions and deliberate choices but by mental processes that are put into motion by features of the environment and that operate outside of conscious awareness and guidance" [11] (p. 462). One of the mechanisms contributing to

such nonconscious influences on attitudes and behavior is implicit memory and its related priming phenomenon.

Rovee-Collier et al. (2001) note the distinction between implicit and explicit memory [12]. Implicit/non-declarative memory is a phylogenetically primitive form of memory thought to become functional early in ontogeny. Implicit memory is automatic, non-effortful, incidental, and fast. These authors note that implicit memories pop into the mind uncontrollably and involuntarily. This type of memory is not associated with conscious awareness of memory retrieval. When one thinks of memory we are typically referring to conscious self-initiated recall or explicit memory. Consequently, the nature of explicit/declarative memory will be reviewed first.

*2.1. Explicit Memory*

Explicit/declarative memory is effortful, deliberate, intentional or voluntary, conscious, and relatively slow [12,13]. The determination of explicit memory is evidenced in free recall, recognition, and cued recall assessments. Tulving noted that explicit memory can be subdivided into two categories: episodic and semantic [14]. "Episodic memory refers to the capacity for recollecting happenings from the past, for remembering events that occurred in particular spatial and temporal contexts. Semantic memory refers to the capacity for recollecting facts and general knowledge about the world" [15] (p. 205).

Regarding explicit memory, episodic memory with its characteristic spatial and temporal contextual features is of primary interest to the present discussion. Dickerson and Eichenbaum note that episodic memories involve "information about unique personal experiences that include time and place of an event as well as detailed information about the event itself" [16]. The hippocampus and parahippocampal areas form the medial temporal lobe (MTL), which is essential for encoding and retrieving episodic memory. Dickerson and Eichenbaum (2010) note that the MTL receives and returns information from virtually all neocortical association areas. Regarding the visual system, the perirhinal cortex receives information from the ventral visual pathway that conveys nonspatial ('what') features that identify stimuli and that the parahippocampal cortex receives spatial ('where') information from the dorsal visual pathway. Both sources of information converge in the hippocampus, where this information is combined into object–context representations [17]. Dickerson and Eichenbaum noted that the hippocampal integration of what, where, and when information is the principal mechanism for episodic memory. These new contextualized representations are then sent back to neocortical areas that initially processed them and stored there for future recollection.

Ritchey et al. (2015) reviewed evidence that the cortical pathways to the hippocampus appear to extend from two large-scale cortical systems: a posterior medial (PM) system and an anterior temporal (AT) system [18]. The posterior medial system includes the posterior cingulate, precuneus, angular gyrus, anterior thalamus, presubiculum, mammillary bodies, and medial prefrontal cortex and connects with the parahippocampal and retrosplenial cortexes. This system is involved with the processing of context information and long-term storage of previously learned contexts. The PM system processes spatial and temporal information and facilitates object–context and object–location associations from a first-person perspective. Moreover, the various brain regions in the PM system have been collectively described as the "default network" and are associated with retrieval of autobiographical events in rich contextual detail.

The anterior temporal system includes the ventral temporal polar cortex, amygdala, and lateral prefrontal cortex, and connects with the perirhinal cortex [18]. The AT system is involved in processing item information and long-term storage of previously learned items. Moreover, this system is involved with learning associations of object features and object–object associations. In addition, the AT system, with its connections to the amygdala and lateral orbital cortex, contributes to item–reward associations critical for decision-making based on past experiences.

In addition to the PM and AT systems converging in the hippocampal formation, Ritchey et al. note that they also connect at the ventromedial prefrontal cortex (vmPFC) [18]. Moreover, these authors believe that this convergence is important for the control of memory-guided behavior. However, for patients with significant trauma histories and consequent PTSD, it is likely that these two systems fail to function adequately to create contextualized autobiographical memories associated with recovery from PTSD.

*2.2. Implicit Memory*

Graf and Schacter (1985) note that "implicit memory is revealed when performance on a task is facilitated in the absence of conscious recollection" [19] (p. 501). Demonstrations of implicit memory are typically independent of MTL activity, and such learning has been demonstrated in severe amnestics deficit in explicit and episodic memory capabilities [20]. Although various types and aspects of implicit memory have been linked to specific cortical and subcortical brain areas, Reber (2013) posits that it represents "a universal principle of plasticity throughout the brain" [21] (p. 2034). In contrast, Schacter (1992) and Squire and Zola (1996) propose that the brain has multiple memory systems and that implicit learning is independent of the hippocampal formation associated with explicit/declarative memory [22,23]. Schott et al. (2005) also report "firm evidence that implicit and explicit memory have distinct functional neuroanatomies" and demonstrated that word priming was associated with decreases in left fusiform gyrus and bilateral frontal and occipital brain activations; in contrast to explicit memory that was associated with bilateral parietal, temporal, and left frontal increases in activation based on hemodynamic responses [24] (p. 1257).

Graf and Schacter (1985) demonstrated that performance on a word completion task was improved for words that the subjects had been previously exposed to when the subjects were instructed to complete word fragments with the first word that came to mind [19]. This phenomenon is referred to as the repetition priming effect, and implicit memory outperforms explicit measures of recall when exposure to the words precludes elaborative/semantic processing. Various strategies are employed to prevent subjects from attending to study materials in ways that would facilitate deliberate learning and foster explicit memory. Moreover, when subjects are required to engage with study material in elaborative ways that improve explicit memory, no further benefit accrues for implicit memory [25].

Slotnick notes that repetition priming is associated with repetition suppression or adaptation [20]. Repetition suppression involves a decrease in brain activation when stimuli are repeated. If subjects are required to make motor responses in subsequent testing situations, reaction times are also faster for repeated items. Overall, this is considered to reflect increased efficiency and more fluent processing in brain activity consistent with adaptive neural plasticity.

Extensive research by Schacter and his colleagues has detailed the specifics of repetition priming, and he has proposed that data-driven priming effects (low level sensory/perceptual aspects of stimuli) reflect the operation of perceptual representation systems (PRS) [23]. These perceptual representation systems are cortically based, operate at a presemantic memory level, and support nonconscious expressions of memory. Moreover, Schacter cites evidence against the notion that priming is mediated by activation of preexisting representations and suggests that the creation of novel representations occurs. Moreover, Schacter (1992) proposes three PRSs: visual word form, structural description, and auditory word form. Camina and Güell (2017) also relate implicit memory to visual and auditory memory systems [26]. Consistent with the PRS model, Gabrieli et al. (1995) described a patient with a right occipital lesion who demonstrated intact explicit memory and impaired visual priming implicit memory [27]. Golby et al. (2005) compared explicit and implicit memory for intentional encoding of scenes in early AD patients and age-matched healthy controls. These authors found that deficits in explicit memory for AD patients was associated with MTL, lingual, and fusiform activity, whereas intact implicit

functioning (visual priming) in AD patients was associated with lateral occipital, parietal, and frontal areas [28].

Reber and others review the neural basis of implicit learning and document various experience-dependent behavioral and cognitive skill acquisitions that are acquired automatically without conscious awareness or explicit recall. This considerable array of automatic learning capacities involving subcortical brain regions supports the role of implicit memory in multiple aspects of trauma related disorders.

**Probabilistic classification learning** involves attempting to learn a set of associations that are not obvious and make predictions based on the stimuli presented. The relationships are difficult to learn and subjects gradually learn to make accurate predictions. One such task involves the subject attempting to make weather forecasts. Both healthy controls and amnestic patients can perform this task, but the amnestic patients are markedly impaired at answering explicit factual questions about the training episode [22]. Squire and Zola (1996) note that nondemented patients with Huntington's or Parkinson's disease, disorders that involve pathology in the caudate nucleus, are impaired with probabilistic learning. Utilizing event-related functional magnetic resonance imaging while healthy subjects engaged in successful weather prediction outcomes was associated with increased activity of the body/tail of the caudate nucleus and the putamen [29]. Immediate and nonconscious recognition of trauma memory triggers based on such classification learning likely contribute to trauma-related fear, avoidant behaviors, and, possibly, dissociative coping responses.

**Motor sequence learning** involves "the experience-based reshaping of activity within the motor and motor-planning cortical areas" [21] (p. 2034). Corkin (1968, 2002) describes her work with H.M., a patient experiencing complete antegrade amnesia and loss of explicit episodic memory due to bilateral resection of his medial temporal lobes [30,31]. She noted that H.M. was able to learn rotary pursuit and mirror tracing skills based on extrahippocampal sites. Reber (2013) noted that "repeatedly executing a motor response sequence produces changes in activity in motor cortex and associated regions of both basal ganglia and cerebellum" [21] (p. 2035). The Serial Reaction Time task involves subjects pressing different keys as indicated by cues presented to them. Within sets of such cues are occasional recurring sequences. Although without conscious recognition of these recurring sequences, subjects evidence reduced reaction times when encountering these recurring sequences. Reber noted that such implicit perceptual-motor sequence learning is associated with reduced activity in cortical areas associated with motor control and task processing along with increased activity in the basal ganglia [21,32]. Gobel et al. (2011) reported impaired perceptual-motor sequence performance in patients with Parkinson's disease, which is associated with basal ganglia dysfunction. Overall, Reber noted that the increased activity of the basal ganglia supports the idea that cortico-striatal connections are important for perceptual-motor sequence learning.

**Statistical learning** involves the ability to acquire structure from relationship regularities distributed in space and time embedded within complex and continuous sensory information in our environments [33]. Such regularities occur in languages and have been considered fundamental to the rapidity of infant language learning. Whereas it was initially thought that such statistical learning was domain-specific, e.g., Chomsky's Language Acquisition Device, multiple studies have documented that such learning in humans occurs with non-language tone sequences and with patterned visual stimuli [33–35]. Moreover, Saffran et al. (1999) have noted that such learning proceeds automatically as a byproduct of mere exposure [34]. In addition, Turk-Brown et al. (2009) found that learning of visual statistical regularities emerged early, involved activation of the striatum and MTL, but that the neural signature of statistical learning was dissociable from subsequent explicit familiarity, suggesting that learning can occur in the absence of awareness [33]. Nonetheless, Ellis et al. (2021) recorded hippocampal activity in infants aged 3–24 months while they viewed sequences of objects and found greater hippocampal activity when the order of the sequence contained regularities that could be learned compared to when the order was

random [36]. Consequently, statistical learning may involve aspects of both implicit and explicit memory.

**Implicit category learning** involves "categorical structures abstracted implicitly from examples based on experience ... and applying this to items not previously seen" [32] (p. 2031). "The implicit category learning system is hypothesized to depend on posterior cortical areas supported by cortico-striatal circuits connecting these to basal ganglia, specifically the caudate" [32] (p. 2032). Gobel et al. (2013) compared implicit perceptual-motor skill learning in patients with mild cognitive impairment (MCI) due to MTL dysfunction and patients with Parkinson's Disease (PD) [37]. As patients with PD have basal ganglia dysfunction that would theoretically interfere with their implicit acquisition of the perceptual-motor skill task, it was predicted that they would not evidence increased performance on trials involving the repeated motor skill sequence. These authors confirmed the importance of basal ganglia in acquisition of implicit perceptual-motor memory as the patients with PD did not evidence improved performance on repeated sequence trials in comparison to non-repeated sequence trials as the healthy controls and mild cognitive impairment patients did. These findings support the critical role of the basal ganglia and cortico-striatal circuits with perceptual and motor planning areas of the cortex. In addition, the statistical structure of the repeating sequences required higher-order associations to be learned among sequence elements, but this did not impair learning in patients with MCI, confirming that this skill was not dependent on MTL explicit memory functions.

**Classical Conditioning** involves the pairing of an unconditional stimulus (UCS) that generates an unconditional response (UCR) with a previously neutral conditioned stimulus (CS) resulting in a CS-UCS association whereby the CS generates the UCR. Rovee-Collier et al. (2001) in their review conclude that both classical and operant conditioning require only a primitive memory system [12]. Clark and Squire (1998) noted that amnestics with hippocampal lesions readily acquire simple delay classical eyeblink conditioning supporting the role of non-explicit learning [38]. Knight et al. (2009) demonstrated that Pavlovian fear conditioning associated with supra and sub-threshold volume auditory tones occurs with and without contingency awareness [39]. Moreover, these authors utilized fMRI to confirm that only contingency awareness trials involved hippocampal and parahippocampal activity whereas conditioned fear was associated with amygdala activity on both perceived and unperceived trials independent of contingency awareness. Additionally, supporting the dual process model of classical fear conditioning, Schultz and Helmstetter (2010) conducted a study of Pavlovian fear conditioning to complex sine wave gratings visual stimuli [40]. Continuously monitoring UCS expectancy and skin conductance responses to "easy" and "difficult" discriminable stimuli, they demonstrated that conditioning can occur on an implicit level without explicit knowledge of contingencies. Confirming the subcortical nature of fear conditioning, LeDoux et al. (1989) found that "visual cortex lesions did not interfere with acquisition, indicating that visual fear conditioning, like auditory fear conditioning, is mediated by subcortical, probably thalamo-amygdala sensory pathways" [41] (p. 238).

### 2.3. Electrophysiological Distinctions between Implicit and Explicit Memory

Another approach to characterizing implicit and explicit memory involves the timing of EEG potentials and their topographies. Event-related potentials (ERPs) can track brain activity and produce temporal resolutions within milliseconds [20]. ERP electrical potentials are typically collected from electrodes covering the scalp that are averaged and reflect summated activity of a large number of positive and negative postsynaptic potentials [42]. These summated potentials or waveforms are time-locked to the onset of a particular stimulus or response and are typically examined from 200 ms prior to the event to 1500 ms following the event. Waveforms can be defined as "changes in the scalp-recorded voltage over time that reflect the sensory, cognitive, affective, or motor processes elicited by a stimulus" [42] (p. 4). Moreover, ERP components that reflect neuronal populations associated with specific information processing can be described by its timing (latency),

change in polarity, and spatial distribution over the scalp. Earlier aspects of waveforms reflect sensory processing, whereas later aspects of waveforms reflect cognitive processes. Whereas ERPs have very high temporal resolutions, their spatial resolutions are much weaker and typically reflect only cortical activity.

The neural correlates of implicit memory/learning have been found to occur earlier in the ERP waveforms, associated with topographies in the parietal areas, and dissociated from explicit memory/learning associated with later and more frontal aspects of waveforms [43–49]. The repetition suppression/priming effect associated with the presentation of repeated stimuli, in comparison to new stimuli, was found to be associated with less voltage negativity in early parietal/posterior waveform components supporting the perceptual representational system model of implicit memory [43,48,50]. Whereas early sensory-based waveform components tend to be negative in polarity, later frontal explicit memory components tend to be positive in polarity [45,48,50]. These studies support the dissociation of neural correlates of implicit and explicit memory systems highlighting that implicit memory is more sensory based.

### 3. Relevance of Implicit Memory to PTSD

From the review of implicit memory above, we see that priming largely occurs in the sensory and associative areas of the cortex (perceptual representation systems), probabilistic classification involves the caudate nucleus and the putamen, statistical learning involves the striatum of the basal ganglia (and probably the hippocampi to some extent), implicit category learning is dependent on the caudate nucleus of the basal ganglia that supports cortico-striatal circuits, and motor sequence learning is dependent on the basal ganglia. With the exception of priming that occurs in the sensory and association areas, all the other forms of implicit learning reviewed involve some portions of the basal ganglia.

Although Brewin et al. (2010) did not specify that his proposed Sensory Representational System that stores the memories that cause intrusive images in PTSD was a form of implicit memory, we can see that this readily corresponds with Schacter's Perceptual Representation System that has been determined to be a form of implicit memory [9,23]. Moreover, using an auditory oddball paradigm adapted to functional magnetic resonance imaging, Bryant et al. (2005) identified greater activity in posterior parietal somatosensory regions for individuals with PTSD compared to age and sex-matched non-traumatized controls [51].

Multiple studies have demonstrated significant priming for trauma-related words, but not for neutral or general threat-related words in subjects with PTSD in comparison to trauma exposed subjects without PTSD. Zeitlin and McNally (1991) studying Vietnam combat veterans found that both veterans with and without PTSD evidenced an explicit memory bias for combat words, but only veterans with PTSD exhibited an implicit memory bias for combat words [52]. Amir et al. (1996) found that Vietnam veterans with PTSD evidenced implicit memory for combat-related sentences, but Vietnam veterans without PTSD did not [53]. However, McNally et al. (1996) subsequently failed to find an enhanced priming effect for combat veterans with PTSD, but speculated that the level of depression in their sample may have interfered with the perceptual priming effect [54]. However, Amir et al. (2010) classified undergraduates based on completion of self-report measures into a PTSD group and three control groups: anxiety, trauma without PTSD, and non-anxious. These authors found that the participants with PTSD symptoms evidenced greater implicit memory on a visual clarity-rating task for negative and trauma-relevant pictures in contrast to the non-PTSD trauma control group, who only showed increase implicit memory for negative pictures [55]. Grégpore et al. (2019) studied sexual abuse victims who did not develop PTSD with non-exposed controls and found that trauma-exposed subjects evidenced increased priming for trauma-related words in comparison to control subjects [56]. Two prospective studies identified that enhanced priming for trauma-related words measured soon after the trauma were predictive of subsequent PTSD development

and severity at three to nine month follow-ups for assault and motor-vehicle accident survivors [57,58].

Kleim et al. (2012) assessed the perceptual processing advantages for trauma-related visual cues in trauma-exposed motor-vehicle accident survivors in a cross-sectional study and assault survivors in a prospective study [59]. Using a blurred picture identification task, these authors found that participants with Acute Stress Disorder or PTSD, but not trauma survivors without these disorders, identified trauma-related pictures, but not general threat pictures, better than neutral pictures. Moreover, this relative processing advantage for trauma-related pictures was correlated with re-experiencing and dissociation, and predicted PTSD at follow-up. Sündermann et al. (2013) had healthy subjects watch neutral or trauma picture stories and found that enhanced perceptual priming predicted intrusive memories at a 2-week follow-up [60]. Concurrent assessment of higher levels of data-driven processing, dissociation, and anxiety during trauma stories also predicted intrusive memories. Minshew and D'Angrea (2015), studying women with histories of chronic interpersonal violence, found them to have both explicit and implicit memory biases for both general threat and trauma-related words, but that only implicit memory for trauma-related words was associated interpersonal sensitivity, hostility, and alexithymia (characteristics they considered to be reflective of complex PTSD in the population they studied) [61].

Kiekorian and Layton (1998) described a case of a healthy 53-year-old man with no prior psychiatric or neurological history who was buried completely under 5.5 m of sand in a construction accident [62]. It required 15 min to uncover him, he was given artificial respiration, and although a CT scan of his brain was normal, he remained in a coma for two days due to the anoxia. An EEG taken two weeks after this anoxic episode was normal. These authors noted that his emotional status was depressed, anxious, and he ruminated continually about sudden death with specific fears of the earth opening up and swallowing him. He had daily nightmares of the same content. However, due to his anoxic encephalopathy, he had no explicit memory for the traumatic event. He was diagnosed with PTSD and underwent four years of psychotherapy focusing on recovery of explicit memory for the accident assuming it had been repressed. The patient consistently maintained that he did not remember the event. This case underscores the role of implicit memory in the development and maintenance of PTSD.

In a meta-analysis of neuroimaging studies, Stark et al. (2015) found that their analysis of studies comparing individuals with PTSD to trauma-exposed controls revealed activation differences in the basal ganglia (bilateral putamen and pallidum, extending to the caudate on the right) and the left fusiform gyrus for individual with PTSD [63]. Taken together, these findings highlight the important role of the implicit memory system in the development of PTSD.

## 4. Peritraumatic Dissociation, Dissociation, and PTSD

Another factor associated with the development of PTSD is peritraumatic dissociation, dissociative reactions during or immediately following a traumatic event. Dissociative reactions often involve perceptual distortions of time, place, or person; disruptions in sensations, perceptions, and memory that are usually well integrated. Dissociative experiences include states of depersonalization, derealization, emotional numbing, and disembodiment. Multiple studies and meta-analyses [64,65] have confirmed the relationship of peritraumatic dissociation to the diagnosis and severity of PTSD symptom severity [66–69]. Although Briere et al. (2005) found that peritraumatic dissociation was associated with development of PTSD, they found in a multivariate analysis including peritraumatic dissociation, persistent trauma-related dissociation, and generalized dissociation, that trauma-related persistent dissociation was a stronger predictor of subsequent PTSD [70]. Although some have questioned the reliability of retrospective reports of peritraumatic dissociation, Daniels et al. (2012) studying an acutely traumatized sample recruited from emergency departments at three time points within the first three months post trauma found that

peritraumatic dissociation predicted PTSD diagnostic status at five to six weeks and three months over and above history of childhood trauma [71]. Moreover, these authors also conducted script-driven traumatic event recall fMRI scans two to four months post trauma and, after controlling for dissociation at the time of the scan, found that peritraumatic dissociation was positively correlated with right lingual and fusiform gyri activation. Noting that these areas are associated with vivid autobiographical memory recall of highly emotional events, they suggested that peritraumatic dissociation directly leads to formation of intrusive memories.

*The Diagnostic and Statistical Manual of Mental Disorders*, *5th edition* (DSM-5) introduced a subtype of PTSD with dissociative symptoms including persistent or recurrent experiences of depersonalization or derealization [1]. Whereas the DSM-5 dissociative criteria are limited to depersonalization and derealization, Ross et al. (2018) identified that individuals with dissociative PTSD also experienced other dissociative symptoms including gaps in awareness and memory and sensory misperceptions [72]. Wolf et al. (2017) developed a self-report measure to specifically assess the presence of the dissociative subtype and via factor analysis identified three dimensions of dissociative symptoms: derealization/depersonalization, loss of awareness, and psychogenic amnesia [73]. In addition, Felmingham et al. (2008) reported that individuals classified as dissociative PTSD reported feeling 'numb', 'unreal', and 'outside of my body' [74].

Multiple studies have documented that the dissociative subtype of PTSD is associated with more severe symptoms of PTSD, more severe memory impairment and self-destructive behavior, more comorbid mental disorders, and having higher treatment needs, greater illness burden, and reduced treatment outcomes [72,75–78]. In addition to the increased severity of symptoms associated with dissociative experiences in PTSD, Boyd et al. (2018) found that these dissociative symptoms were the strongest correlate of functional impairment and that dissociative and derealization symptoms significantly mediated the relation between PTSD symptoms and functional impairment [79].

One of the possible mechanisms of the relationship of dissociative symptoms in PTSD and the increases in PTSD symptom severity, memory disruption, and functional impairment may be related to the way fear is processed in dissociative PTSD. Felmingham et al. (2008) monitored fMRI brain activation to fearful versus neutral facial expression during conscious and non-conscious (backward masking) conditions and found that for dissociative PTSD subjects, in comparison to the non-dissociative PTSD subjects, the non-conscious presentations enhanced activation in the bilateral amygdala, insula, and left thalamus [74]. They concluded that activation of basal ganglia (pallidum) suggested a pattern of enhanced early sensory registration, somatosensory arousal, and motor readiness that is consistent with enhanced automatic arousal in the dissociative group. Brown and Morey (2012) reviewed neural systems for cognitive and emotional processing in PTSD and identified a *threat-alerting* component that consists of the amygdala, insula, and vmPFC and a *threat-assessing* component that consists of the hippocampus, anterior cingulate, striatum, dorsomedial PFC, precuneus, and ventrolateral PFC [80]. These authors indicated that the threat-assessing component that involves frontal and default mode neural sites (associated with autobiographical information processing) is ineffective in PTSD and fails to activate top-down emotional modulation mechanisms. Taken together, these studies support the subcortical activation of fear and impaired cortical inhibition of subcortical structures in dissociative forms of PTSD.

*4.1. Dissociation and Memory*

Van der Kolk (1994) noted that "trauma is stored in somatic memory", that in PTSD there is a "failure of declarative memory", and that "intense emotional memories are processed outside of the hippocampally mediated memory system and are difficult to extinguish" [81] (p. 253). This model is consistent with the dual representation theory of Brewin et al. (2010) and the cognitive model proposed by Ehlers and Clark (2000) [2,9]. Michael et al. (2005) found that distress caused by intrusive recollections, their "here and

now" quality, and their lack of context predicted PTSD severity; whereas the presence and frequency of intrusive recollection together account for only 17% of the variance in PTSD severity, the nowness, distress, and lack of context explained an additional 43% of the variance [82]. Nash et al. (2014) have also described peritraumatic amnesia as a failure to encode episodic memories near the time of the trauma [83].

More recent studies have confirmed the association of dissociative symptoms and explicit/declarative memory impairments. McKinnon et al. (2016) found dissociative symptoms to be associated with impairments in measures of attention and memory; Parlar et al. (2016) found greater severity of derealization to be related to reduced performance on a test of verbal memory recognition [78,84]. Moreover, Carrion et al. (2010) found that adolescents with PTSD secondary to interpersonal trauma, in comparison to age and gender-matched healthy controls, evidenced decreased verbal declarative memory performance and reduced right hippocampal activity during the retrieval component of the task [85]. In addition, these authors found that reduced left hippocampal activity during retrieval was correlated with numbing (a dissociative experience) and avoidance symptoms. Allen et al. (1999) have also reviewed the association of dissociative detachment encoding disruptions and impaired autobiographical memories [86].

It is not unusual that states of helplessness during traumatic events are associated with dissociative experiences that involve a sense of disembodiment and analgesia. Trauma survivors often report viewing the event from a third person perspective, e.g., floating above one's physical self and viewing the event from a disembodied, out-of-body, perspective. Bergouignan et al. (2014) noted "long-term episodic memory impairments in psychiatric conditions with dissociative symptoms, in which individuals feel detached from themselves as if having an out-of-body experience" [87] (p. 4421). These authors hypothesized that the capacity for successful encoding of autobiographical episodic memory was dependent on the experience of the world from the perspective of one's own body. They used a "multisensory full-body illusion to move the center of the bodily and spatial awareness (sense of bodily self) from the location of the real body to the other end of the testing room such that the test individual experienced the life event from outside her/his body" (p. 4422). When comparing the viewing of a real-life event (portrayed by an professional actor) from an out-of-body perspective to subjects experiencing it from the sense of bodily self colocalized with the real body, these authors found an episodic recollection deficit specifically associated with activity changes in the left posterior hippocampus. Specifically, they found reduced activity in the left posterior hippocampus during initial retrieval and increasing activity with subsequent retrievals. In contrast, for subjects viewing the event from an in-body perspective, left posterior hippocampal activity was strongly activated on initial retrieval, but showed progressively less engagement with further repetition (a rehearsal effect). Moreover, activation of the left posterior hippocampus during repeated retrievals was negatively correlated with subject ratings of vividness of remembered events encoded out-of-body compared with in-body encoding. Taken together, these researchers concluded that "episodic encoding of life events requires perceiving the world from the first-person perspective centered on one's real body" and that "encoding events experienced out-of-body specifically impact the activation of the left posterior hippocampus during retrieval, suggesting an impaired hippocampal binding mechanism during encoding" (p. 4424).

### 4.2. Contributing Factors to Dissociation and PTSD

Both Schore (2003) and Liotti (2004, 2006) have related childhood abuse and neglect, disorganized attachment, and propensity for dissociation to the development of PTSD [88–90]. Specifically, when children have a parent(s) with unresolved trauma that relates to them with frightened/frightening states, these children are unable to develop an organized attachment to them. Faced with their biological attachment motivation to seek primary caregiver proximity and support when distressed and then confronted with frightened/frightening parental behavior, they experience simultaneous approach and

avoidant impulses. Hesse and Main (2006) have described this situation as "fright without solution" and that it prevents the child from developing an organized form of attachment that either consistently inhibits (Avoidant/Dismissive attachment) or intensifies (Resistant/Preoccupied attachment) the attachment system [91,92]. Infants with disorganized attachment evidence behaviors similar to dissociative symptoms in adults such as disorganized/disoriented behaviors, dazed and trancelike expressions, and simultaneously or rapid sequential contradictory actions. Confronted with an irreconcilable conflict, dissociation or compartmentalization occurs. Consequently, as noted above, Shore and Liotti have suggested that this predisposition to dissociate contributes to peritraumatic dissociation and the development of PTSD or other dissociative conditions when these infants experience subsequent traumatic events.

Supporting the relationship of disorganized attachment and dissociation, Paetzold and Rholes (2021) found that a self-report measure of disorganized attachment in adults was strongly correlated, 0.68, $p < 0.001$, with current dissociation [93]. Van Ijzendoorn et al. (1999) conducted a meta-analysis of studies of infant attachment classifications and found modest short- and long-term stability of disorganized attachment and a tendency of disorganized infants to show dissociative behaviors later in life [94]. In a review article, Lyons-Ruth et al. (2006) cites Barach (1991) as the first theorist to relate dissociation and attachment, suggesting that Multiple Personality Disorder (Dissociative Identity Disorder in the current nomenclature) was a variant of an attachment disorder. These authors also review other research supporting the relationship of disorganized attachment and dissociation [95,96]. Ogawa et al. (1997) also found support for Liotti's theory that disorganized attachment and subsequent trauma are associated with dissociation in adulthood [97,98]. Moreover, Bakermans-Kranenburg and van IJzendoorn (2009), in a review of 205 studies, found that clinical subjects showed more insecure and unresolved attachment representations (unresolved refers to unresolved/disorganized, disoriented classifications that are related to disorganized) than the normal groups, that adults with abuse experiences or PTSD were mostly unresolved, and that "unresolved loss or trauma as assessed with the AAI is an almost perfect marker for dissociative disorders like PTSD, which sheds light on the etiology and mechanisms of these disorders as (partly) attachment disorders" [99] (p. 249). Hébert et al. (2020) studied a sample of 424 preschool children and found that disorganized attachment and emotional dysregulation mediated the association between child sexual abuse and dissociation; specifically, they found that childhood sexual abuse was associated with greater disorganization scores, that were associated with higher emotional dysregulation scores, that were associated with increased severity of dissociation [100]. Studying a sample of at-risk women, Bailey et al. (2007) found that "women classified as unresolved reported higher levels of dissociation, confusion regarding self-identity, and relationship problems" [101]. Anderson and Alexander (1996) found that for adult women survivors of incest, 51% were found to have fearful-avoidant attachment (similar to AAI classification of disorganized) and that fearful-avoidant attachment was the strongest predictor ($p < 0.006$) of dissociation in a simultaneous regression along with three other attachment classifications [102]. Farina et al. (2013) found that 93% of their sample of subjects with dissociative disorders had AAI classified unresolved disorganized attachments [103]. Thomas and Jaque (2014), examining the relationships among AAI unresolved attachment, non-pathological dissociation (Dissociative Experience Scale; DES-Absorption), pathological dissociation (DES-Taxon scores [104]), and supernatural beliefs, found in a multiple analysis of variance that individuals classified as unresolved versus non-unresolved attachment scored significantly higher on both non-pathological and pathological dissociation [104,105]. In contrast, these same authors in a prior study of artists and performing artists found that unresolved classifications were associated with increased pathological dissociation, but not normative dissociation (absorption and imagination) [106]. Taken together, these finding strongly support the relationship between disorganized forms of attachment and dissociative experiences, as well as Loitti's hypothesis that disorganized attachment contributes to the subsequent development of dissociative disorders.

*4.3. Disorganized Attachment and PTSD*

In addition to the specific role of disorganized attachment increasing dissociation as a contributor to the development of PTSD, multiple studies have documented the relationship of disorganized attachment and insecure attachments, in general, to the diagnosis and severity of posttraumatic symptoms [107]. For instance, Mikulincer et al. (2015) in a literature review reported that attachment insecurities were associated with PTSD severity and that attachment security had a healing effect on these symptoms [108]. Van Hoof (2019) found that increased incidence of disorganized attachment as well as unresolved loss differentiated subjects with histories of childhood sexual abuse related PTSD from adolescents with anxiety/depression and controls without psychiatric symptoms [109]. Comparing attachment disorganization in different clinical groups, Juen et al. (2013) found the highest proportion of attachment disorganization in patients with PTSD and Borderline Personality Disorder, 83.3% and 76.5%, respectively. Moreover, they reported that PTSD patients were emotionally overwhelmed by projective attachment scenes compared to the other clinical groups [110]. Furthermore, Juen et al. (2014), examining individuals following an accident-related physical injury requiring inpatient hospitalization, found that individuals with disorganized attachment experienced higher levels of posttraumatic stress symptoms immediately after the accident [111]. In a meta-analysis of adult attachment styles and PTSD symptoms, Woodhouse et al. (2015) found that attachment categories associated with higher levels of anxiety were most strongly associated with PTSD symptoms and that the fearful attachment (most similar to AAI disorganized category) displayed the strongest association [112]. Independent of identification of degree of dissociation, we see that indices of disorganized attachment alone are often seen in cases of PTSD and associated with increased PTSD symptomatology.

Sensory-based emotional intrusions characteristic of PTSD have been related to failure of frontal brain sites to adequately inhibit amygdala activity both at the time of traumatization and during subsequent triggering, a recent study by van Hoof et al. (2019) found a possible neurophysiological basis for the relationship between disorganized attachment and PTSD symptoms [109]. These authors assessed the relationship between Ud (unresolved loss or trauma—a nonconscious indicator of disorganized attachment on the AAI) and amygdala functional connectivity with various brain sites in a mixed group of adolescents with child sexual abuse-related PTSD, anxiety and depressive disorders, and without psychiatric disorders. After controlling for the general psychopathology factor, age, gender, pubertal status, and IQ, these authors found that individuals with higher levels of Ud showed stronger left amygdala connectivity with the lateral occipital cortex, precuneus, and left superior parietal cortex compared to individuals with lower levels of Ud. In addition, Ud was negatively associated with left amygdala–medial frontal cortex connectivity. These stronger functional connections between the left amygdala and occipital and parietal cortexes, and precuneus are consistent with bottom up intrusive visual and emotional activations involving the superior parietal areas, amygdala, and precuneus proposed by Brewin et al. [9]. Moreover, the decreased functional connectivity between the left amygdala and the medial frontal cortex is consistent with the failure of PFC inhibition of amygdala activation, increased fear conditioning, and decreased fear extinction [113–115]. These findings support interventions to modify disorganized attachment styles to both decreased PTSD symptomatology and foster better integration of implicit and explicit memory systems.

## 5. Attachment and Implicit Memory

Bowlby (1969/1983, 1992) outlined the nature and importance of the infant's attachment to his/her mother or primary caretaker [116,117]. Based on the primary caregiver's response to the infant's attachment behaviors, the child comes to develop an Internal Working Model (IWM) for expectations regarding interactions with the social environment and one's ability to influence it [92]. These IWMs can be reliably identified by the age of 18 months old as patterns of reactions to the separation and reunion of the child to her/his

primary caretaker (via the Strange Situation paradigm developed by Ainsworth et al., 1978) [118]. These IWMs are acquired prior to the development of language and the full capacity of the declarative memory system [119,120] and are considered strongly related to implicit memory mechanisms. Mikulincer and Shaver (2017) have noted that IWMs of these parental interactions "become part of a person's implicit procedural knowledge, tend to operate automatically and unconsciously, and are resistant to change" [92]. Similarly, Cortina and Liotti (2007) noted that "memories of the first four years of life are not usually available for recall in verbal narrative form that is the hallmark of autobiographical memory" but "nonetheless, early experience is carried forward in the form of nonconscious, automatic expectations and attributions" [121] (p. 43). Several others have also noted the likelihood that these early representations involve the implicit memory system [122–125]. Attachments also develop later in life when the explicit memory system is accessible. In this respect, Galynker et al. (2012) examined the neural networks subserved by early and later formed attachments [126]. These authors found that the effects of attachment were in the cortico-striato-thalamic circuits and employed a subtraction methodology and multiple linear regressions to determine distinct correlations with a measure of attachment security (Coherence of Mind from the Adult Attachment Interview, AAI) [127]. Comparing neuronal activity of exposure to pictures of early, mother, and late, friend, attachment figures during fMRI, these researchers found that subtraction of friend from mother scans (early attachment) revealed only sub-cortical activations (right medial thalamus and left ventral caudate) with a strong linear relationship between relative activations and AAI scores. In contrast, for the neural activity associated with subtraction of the stranger from the friend scan (late attachment), no attachment-related activity was found in the sub-cortical areas. These finding support the prominent role of implicit memory in the development of critical early life attachments.

As these IWMs of attachment influence one's behavior in attachment relationships and often interpersonal behavior in general, it is interesting to note that the basal ganglia have been considered to influence behavior selection [128,129]. In their review of motivated behavior, Da Cunha et al. (2012) note that "different actions can be selected in the striatum" and that "the basal ganglia is thought to play a role in the actions-selection processes needed for expression of both declarative and procedural memory" [130] (p. 747). Furthermore, Stephenson-Jones et al. (2011) concluded that the mammalian basal ganglia have been co-opted for multiple functions over the course of evolution allowing them to process cognitive, emotional, and motor information in parallel and control a broader range of behaviors [131] (p. 1081). Consequently, the implicit subcortical nature of IWMs involving the basal ganglia is consistent with the control of attachment-related behavior. This may also be a factor related to the difficulty changing such behaviors with top-down cognitive psychotherapy approaches and the need for specialized interventions that can modify these subcortical implicit memory representations.

### 5.1. Positive Effects of Secure Attachment Priming

As noted above, Mikulincer et al. (2015) indicated that attachment security had a healing effect on PTSD symptoms [108]. In this regard, Selcuk et al. (2012) found that activation of a mental representation of an attachment figure (vs. an acquaintance or stranger) following recall of an upsetting autobiographical memory enhanced recovery, eliminated the negative effects of memory recall, and negative thinking in a subsequent stream of consciousness task [132]. Moreover, the degree of recovery predicted mental and physical health in daily life.

Two studies have demonstrated that activation of attachment representations can influence memory consolidation and reduce unpleasant recollections and intrusions. Bryant and Chan (2017) had subjects view negative or neutral images and two days later received an attachment or control prime immediately prior to free recall of the images. An additional two days later, participants reported how frequently they experienced intrusions of negative images. The attachment primed group reported less distress and fewer intrusions than the

control group [133]. Bryant and Foord (2016) had subjects view traumatic or neutral images preceded by subliminal attachment-related or non-attachment-related primes. Individuals with low avoidant attachment styles who received the attachment primes recalled fewer traumatic memories and reported fewer intrusions. Even individuals with high anxious attachment styles reported fewer memories when they had attachment primes [134].

In a systematic review of 30 studies of attachment security priming, Rowe et al. (2020) concluded that security priming improved positive affect and reduced negative affect relative to control primes. They also noted that supraliminal and subliminal primes were equally effective in enhancing security and that repeated priming studies showed a cumulative positive effect of security primes over time [135]. McGuire et al. (2018) found that security primes in both laboratory and in naturalistic settings were associated with greater decreases in depressive symptoms [136].

In another systematic review, Gillath and Karantzas (2019) reported that "security primes activate a sense of attachment security by making mental representations in one's memory more accessible and salient" [137] (p. 86). These authors found that supraliminally administered security priming was associated with beneficial effects across diverse domains including affect and emotional-wellbeing, prejudice and hostile attitudes, empathy-related processes, increased comfort in romantic relationships, and emotional regulatory processes. Moreover, they reported that "the findings speak to the effectiveness of a specific kind of security priming—guided imagery or visualization of a security enhancing interaction" (p. 93). Interested in the impact of repeated attachment security priming on relationship- and self-views, Carnelley and Rowe (2007) found that attachment compared to neutral primes over three days was associated two days later with more positive relationship expectations, more positive self-views, and less attachment anxiety than assessments prior to the primes [138]. Moreover, Gillath et al. (2008) reviewed studies of repeated priming of secure attachment and concluded that studies conducted thus far "strongly suggest that repeated security priming can have persisting effects" [139] (p. 1662).

Priming with secure attachment stimuli tends to be more effective with individuals low on attachment avoidance [132–134,140–142]. Individuals with avoidant or dismissive attachment styles have experienced relatively consistent primary caregiver rejection and emotional neglect. Consequently, they develop a protective pattern that turns off the behavioral attachment system and shun away from attachment opportunities anticipating negative outcomes. Consequently, it is not surprising that they tend to dismiss secure attachment primes and are less likely to experience benefits as do securely attached individuals or even those with anxious attachment patterns that tend hyperactivate their attachment system. However, in general, higher levels of both anxious and dismissive attachment styles are associated with greater threat response and benefit less from attachment primes than individuals with secure attachment styles [143].

### 5.2. Neural Mechanisms of Secure Attachment Priming

Although it is a premise of this paper that the ultimate, stable, and beneficial changes occur in working models is at the implicit subcortical level, several studies have demonstrated cortical emotional regulation benefits of increasing secure attachment-related neural activity. Karremans et al. (2011) noted that distress due to social exclusion was associated with activation in the lateral and medial prefrontal cortex, ventral anterior cingulate cortex, and hypothalamus (potentially leading to HPA stress-related secretion of glucocorticoids). When participants imagined the presence of a secure attachment figure versus a non-attachment control figure during a gaming social exclusion condition, they evidenced less activation in these distress and stress-related brain areas [144]. Assessing the underlying neural mechanisms of increases in the sense of safety and security generated by representations of positive attachment figures, Eisenberger et al. (2011) found that viewing pictures of long-term romantic partners, in comparison to control pictures, while receiving painful stimuli led to reductions in self-reported pain and less pain-related neural activity in the dorsal anterior cingulate cortex and anterior insula along with increased activity in the

vmPFC (a brain area known for inhibiting amygdala activity). Moreover, greater vmPFC activity in response to partner pictures was associated with longer relationship lengths and greater perceived partner support [145].

Norman et al. (2015) found that individuals exposed to threatening faces exhibited amygdala activation, but that subjects experiencing supraliminal attachment security priming pictures prior to threatening exposure did not [146]. Consistent with lower activation of amygdala activity in the presence of an attachment prime, Toumbelekis et al. (2018) found that thinking about a supportive attachment figure in comparison to a positive non-attachment experience prior to fear conditioning was related to reduced acquisition of fear-potentiated startle and level of fear two days later during an extinction recall task [147]. Toumbelekis et al. (2021) also found that imaging an attachment figure in comparison to a non-attachment positive experience immediately following fear conditioning was associated with significantly lower levels of fear recall at both physiological and subjective levels two days later [142]. Assessing the impact of an attachment versus a positive non-attachment prime on memory reconsolidation, Bryant and Datta (2019) had subjects view a traumatic film and recall the film the next day before presentation of the prime. These authors found that exposure to the attachment prime was associated with reduced vividness and distress of intrusions over the next week [141]. Thus, activation of secure attachment representations appears to buffer the impact of threat at both cortical and subcortical levels. Norman et al., agreeing with Coan (2010) [148], reported that "attachment security regulates threat-reactivity through bottom-up modulation of threat appraisal processes, rather than top-down prefrontal mediated regulation" [146] (p. 837). This bottom-up influence of attachment primes is consistent with secure (and other attachment IWMs) being activated at subcortical levels.

Priming with an attachment figure was also found to be associated with higher levels of heart rate variability, a noninvasive measure of ventral vagal parasympathetic activity, during a cold pressor test for subjects with low avoidant attachment levels by Bryant and Hutanamon (2018) [140]. Baldwin et al. (2020) found that individuals with higher levels of anxious and avoidant attachment styles were more likely to evidence a threat response as indexed by decreases in heart rate variability following an initial compassion-focused guided imagery experience. However, following an attachment security prime, heart rate variability increased during a second compassion-focused imagery experience [143]. Assessing heart rate variability (HRV) in the context of a social isolation experience, Liddell and Courtney (2018) found that secure attachment priming was associated with maintenance of HRV in comparison to non-attachment priming that was associated with fluctuations in HRV [149]. These findings indicate that attachment priming increases or maintains HRV and its physical and emotional benefits via increasing ventral vagal parasympathetic activity.

Porges (2011, 2021) has noted that neuroception of safety is necessary for activation of the ventral vagal complex and its related increase in HRV [150,151]. Moreover, Porges (2011) has documented that the nervous system is continuously processing sensory information to evaluate risk and that the "neural evaluation of risk does not require conscious awareness and may involve subcortical limbic structures" (p. 57). Furthermore, he states that the term "*neuroception* was introduced to emphasize a neural process distinct from perception that is capably of distinguishing environmental (and visceral) features that are safe, dangerous, or life-threatening" [150] (p. 58). He explains that when an individual experiences a sense of safety, the activity of the VVC supports a state of social engagement marked by positive emotional facial expressions and prosodic speech that communicate to others that the individual is non-threatening. Moreover, these positive emotional expressions and prosodic speech generate a sense of safety in others that fosters mutual social engagement, cooperation, and attachment. These social engagement and affiliative behavior offers a survival advantage. Furthermore, the sensitivity of neuroception to social communications such as prosody of speech and positive emotional facial expressions is consistent with the positive impact of secure attachment priming.

Porges (2011) sites the work of Morris et al. (1999) that demonstrated increased subcortical connectivity between the right amygdala, pulvinar, and superior colliculus when subjects were exposed to masked fear-conditioned unseen faces rather than seen faces [152]. Right amygdala connectivity with the fusiform and orbitofrontal cortices decreased in the same condition. "These results suggest that a subcortical pathway to the right amygdala, via midbrain and thalamus, provides a route for processing behaviorally relevant unseen visual events in parallel to a cortical route necessary for conscious identification". [152] (p. 1680). More recently, Garrido et al. (2012) documented a dual model of input to the amygdala of behaviorally important sensory information [153]. Using auditory stimuli, these authors found that the subcortical pathway was most important early in stimulus processing and that this "expedited evaluation of sensory input is best explained by an architecture that involves a subcortical path to the amygdala" (p. 129). Here, again we see the importance of subcortical, nonconscious processing of sensory threat-related stimuli in the activation of fear.

In summary, we see that activation and deactivation of a primary neural site for fear conditioning, the amygdala, is modulated by subcortical and cortical activity. That dissociation, a factor associated with the development and severity of PTSD, is associated with early and subsequent states of disorganized and insecure attachments. In contrast, secure attachment priming is associated with decreased fear conditioning, constructive modification of fear memories, and decreases in intrusive recollections. Moreover, as noted above, comparison of neural activity associated with PTSD in comparison to trauma-exposed controls involves the basal ganglia, the same brain region associated with early attachments. Furthermore, emotional and physical health-related ventral vagal complex parasympathetic nervous system activity is a function of nonconscious neuroception of interpersonal safety at a subcortical level that is enhanced by activation of secure attachment representations that likely involve subcortical activity. All these findings support the role of subcortical processes and implicit learning to the development and perpetuation of PTSD.

## 6. Modification of Implicit Memory and Trauma-Related Recovery

As noted above, the implicit memory system involves automatic, nonconscious processes that extract information and relationship of external stimuli from the environment on a continuous basis. As with fear conditioning, it also associates internal and external information with physiological, emotional, and behavioral responses. Such learning does not require conscious awareness to occur. In addition, subliminal and supraliminal priming has been demonstrated to influence implicit memory aspects of PTSD symptoms, consolidation and reconsolidation of fear conditioning, attachment related behavior. Moreover, it has been noted that visual guided imagery of secure attachment representations has been one of the most effective priming strategies and that repeated priming results in persistent effects.

Modification of memory has been the focus of several interventions related to recovery from PTSD and dissociative disorders. Ehlers and Clark (2000) and Brewin et al. (2010) have underscored the importance of integrating contextual information into sensory representations of traumatic experiences as well as reconsolidating memory updates as a means of reducing traumatic intrusions and reliving symptoms of PTSD [4,9]. Meyerson (2010) has recommended specific memory modification and memory creation methods in hypnotic psychotherapy for recovery from traumatic experiences and adverse early childhood experiences [154].

Barach (1991), Liotti (1992, 2004, and 2006), Schore (2002), and Cortina and Liotti (2007) have noted the relevance of disorganized attachment to the development of dissociative and trauma-related disorders [89,90,96,98,121,155]. Moreover, these authors underscore the importance of taking a phase-oriented treatment approach focusing on repair of the attachment disorder first as doing so is associated with decreases in dissociation and establishment of affective stabilization and mental organization that supports higher-order functions and recovery. Recognizing the importance of affecting attachment repair for the

treatment of complex PTSD, dissociative disorders, and developmental trauma, Brown and Elliott (2016) published their Three Pillar approach to comprehensive attachment repair [156]. The primary intervention in their approach involves the co-creation of interactive imaginal scenes of the client as a young child interacting with Ideal Parent Figures engaging with them in ways supportive of secure attachments. In contrast to trauma-focused interventions that involve recollecting traumatic events, Brown and Elliott's approach involves only positive images and consequent positive affects related to desirable behaviors of secure attachment experiences. Co-creating these imaginal experiences on a repeated basis allows for the implicit memory to internalize new information, what the authors refer to as remapping the internal working models.

Brown and Elliott (2016) reported on a pilot study of 12 patients who all started treatment with AAI severe disorganized insecure attachments and ended treatment with attained earned secure status on the AAI. Moreover, their average coherence score on the AAI, a primary indicator of disorganized attachment, from pre- to posttreatment assessment significantly increased from a mean of 2.21 to 7.91. Although not addressed in this pilot study, the authors reported on a case of a woman with dissociative identity disorder, AAI classification of CC (Cannot Classify) with unresolved Ud status indicative of disorganized attachment with fear-based anxious preoccupation and dismissing features but signs of some valuing of attachment. At the conclusion of treatment, this patient evidenced resolution of her unresolved status, decreases in dissociative symptoms to the point that she no longer met the criteria for dissociative identity disorder, and reduction in depressive symptomatology from the moderate to severe range to asymptomatic range on the Beck Depression Inventory. Brown and Elliott (2016) noted that "the most remarkable finding illustrated by this case is the clinically significant drop in dissociative, traumatic, and depressive symptoms and disorders from treatment that was *entirely attachment-based* . . . . We believe that increased organization of the mind per se has a positive treatment effect on trauma-related symptoms independent of trauma processing" [156] (p. 611).

The benefits of the interventions just described were accrued over treatment intervals of one to 3.5 years, and it is understandable that attainment of such changes would not be the product of short-term interventions. However, Parra et al. (2017) utilized only the Ideal Parent Figure (IPF) protocol over four weekly sessions in the treatment of patients with severe complex PTSD related to childhood trauma [157]. These researchers found that use of co-created generic IPF imagery sessions recorded for participants to practice between sessions was associated with significant decreases in symptom severity and attachment traumatization along with increases in quality of life from pretreatment levels to 1-week and 8-month post-treatment assessments. Moreover, at the 8-month assessment, participants reported continued use of the recording and use of them following episodes of emotional distress to facilitate recovery. Taken together, the work of Brown and Elliott (2016) and Parra et al. (2017), along with the attachment priming studies noted above, suggest that repeated imaginal exposure of oneself as a child experiencing secure attachment interactive scenes allows the implicit memory system to internalize such experiences and modify IWMs. Moreover, this occurs automatically at a nonconscious level.

## 7. Discussion

Ehler and Clark (2000) and Brewin et al. (2010) have both demonstrated that PTSD involves disruption in memory processes; specifically, a failure to integrate sensory and fear-related memories with explicit autobiographical memories [4,9]. With their emphasis on information processing, both of these models underscore the importance of integrating contextual information of the explicit memory system with sensory-fear related memories. Brewin et al. have also suggested memory reconsolidation strategies that weaken sensory-fear related memories. Disruptions in the hippocampal-based episodic memory system at the time of traumatization is a likely factor contributing to inadequate memory integration.

The goal of fostering memory integration as a mechanism of PTSD recovery often focuses on modification of trauma memories with an emphasis on the explicit autobiograph-

ical memory system. Two frequently employed empirically supported techniques are prolonged exposure (PE) and eye-movement desensitization and processing (EMDR) [158,159]. Whereas both PE and EMDR attribute treatment efficacy to memory integration, only PE emphasizes the role of habituation, a form of non-associative implicit memory, as a mechanism of change. Nonetheless, all of the information processing models of treatment focus on elaboration of conscious autobiographical memory. Moreover, to this author's knowledge, none of these trauma-focused methods consider building positive implicit memories alone, i.e., IWMs, as a mechanism of change.

Another consistent finding with the application of any specific form of treatment for PTSD is that not everyone benefits from it. Reviewing reduction in PTSD symptomatology in a sample of 2715 Veterans following VA residential PTSD treatment, Sripada et al. (2019) found that although "65% of the sample exhibited a decrease in Posttraumatic Checklist scores, only 36% experienced clinically significant improvement" [160] (p. 21). Moreover, Schottenbauer et al. (2008) reported that rates of non-responders for exposure therapy/prolonged exposure ranged from 20% to 50% and for EMDR ranged from 7.3% to 92% [161]. Cognitive Processing Therapy (CPT) is another empirically supported intervention for PTSD that also involves review of autobiographical memories and cognitions related to traumatic experiences [162]. Resick et al. (2021), reporting on the outcomes of CPT, noted that "nearly on quarter of participants (24%) did not reach a good end-state of PTSD symptoms as defined by a score of less than 20 on the PCL-5 after 24 sessions or after having completed 18-weeks of treatment" [163] (p. 6).

Findings like these, along with the appreciation of dissociative symptoms, disruptions in identity, and skill deficits, have led Gold (2000, 2020) and Liotti (1992) to speculate that trauma-focused interventions alone are inadequate to facilitate recovery from the experience of extensive trauma. One of the factors these authors address is repair of disorganized attachment.

As noted above, the majority of empirically supported interventions for PTSD focus on recollection and elaboration of the autobiographical trauma memories. However, this intervention alone has not been found to consistently resolve PTSD. In fact, Bae (2016) found in their final logistic regression of EMDR PTSD non-responders that dissociation and the presence of two or more comorbidities (as is quite common with complex PTSD or dissociative disorders) were the significant variables [164]. Moreover, it is the premise of this paper that such interventions are not fully effective because they do not adequately address the representations of trauma in the implicit memory system. In addition to the sensory-fear memory associations that likely involve subcortical implicit learning, basal ganglia activity was found to be associated with early attachment representations as well as PTSD (when isolated from trauma exposed individuals without PTSD).

Multiple studies have found that PTSD, dissociative forms of PTSD, and complex forms of PTSD are more likely to occur when the trauma is of an interpersonal nature rather than a non-interpersonal nature [165–167]. The therapeutic context of trauma recovery work emphasizes the role of the therapist–patient relationship and the importance of the therapist demonstrating good enough secure attachment behaviors in the therapeutic relationship [156,168]. Moreover, Porges (2011) notes that neuroception of safety "might be triggered by feature detectors involving areas of the temporal cortex that communicate with the central nucleus of the amygdala and the periaqueductal gray, since limbic reactivity is modulated by temporal cortex responses to the intention of voices, faces, and hand movements. Thus, the neuroception of familiar individuals and individual with appropriately prosodic voices and warm, expressive faces translates into a social interaction promoting a sense of safety" [150] (p. 58). As the neuroception of safety and consequent activation of the ventral vagal complex has been predicted to promote improved gastrointestinal function, Damis and Hamilton (2020) demonstrated that repeated hypnotic imagination of scenes that generate a sense of safety for the patient resulted in increases in experienced sense of subjective safety and decreases in symptoms of brain–gut interaction disorders (functional gastrointestinal disorders) [169]. This study provides evidence that patient

identified scenes of safety can increase ventral vagal activity, similar to scenes of secure attachment primes, and that the mechanism of action likely involves subcortical processes.

Given the sensitivity of the neuroception of safety to interpersonal behaviors (e.g., prosody of voice, warm emotionally expressive faces), it is no surprise the secure attachment primes are a protective influence on HRV in the face of threat (e.g., social isolation, pain) [150]. Moreover, it is likely that this benefit is accrued through subcortical implicit mechanisms. In addition, as seen in the application of the repeated imagery of ideal parent figures, imagery of secure attachment interactions allows the implicit memory system to internalize new internal working models, or remap them as Brown and Elliott (2016) propose [156]. Furthermore, it is this authors opinion that internalization of a sense of safety via safe-place and secure attachment imagery should be established early in treatment to set the neurophysiological substrate (VVC activation) for attachment repair.

In conclusion, it appears that will, conscious intention, or activation of conscious memory is not always sufficient to resolve PTSD and that creating the circumstances (imaginal scenes) for the implicit, automatic, nonconscious memory system to modify its pathological representations may be necessary for recovery from traumatizing experiences.

Approaches to optimize neural networks that include modification of autobiographical memories are also very promising. Frewen and Lanius (2015) have proposed a Four-Dimensional Model of the Traumatized Self identifying dimensions of time, thought, body, and emotion [170]. These authors note that symptoms in these categories move from states of normal waking consciousness to trauma-related states of consciousness associated with greater degrees of dissociation as symptom severity increases. They note that the default mode network, an important network for autobiographical memory, is disrupted in patients with PTSD and site work by Blum et al. (2009) demonstrating a lack of functional connectivity between the anterior hub (which includes the medial PFC) and the posterior hub (the posterior cingulate cortex and the precuneus) [171]. Others have also found disruptions in DMN connectivity in patients with PTSD [172,173]. Daniels et al. (2011) noted that deficit DMN connectivity in patients with childhood maltreatment-related PTSD was similar to DMN connectivity observed in healthy children aged seven to nine years and speculated that early life trauma may interfere with the developmental trajectory of the DMN and its functions [174].

Neurofeedback, a form of biofeedback focusing on modification of EEG activity, has demonstrated effectiveness in the treatment of PTSD [175–177]. Moreover, Kluetsch et al. (2014) found that alpha desynchronizing neurofeedback was associated with enhanced DMN connectivity involving the bilateral posterior cingulate cortices and the right middle frontal gyrus and the left medial PFC [178]. Consequently, neurofeedback interventions that modulate brain activity and connectivity offering another strategy for the treatment of PTSD.

It has also been noted that activity dependent neuroplasticity occurs in the hippocampus. Maguire et al. (2006) and Woollett and Maquire (2011) have documented increases in gray matter mid-posterior and posterior hippocampi volume in taxi drivers associated with increased spatial knowledge of the large complex city of London [179,180]. These findings, along with the benefits of increasing hippocampal-dependent spatial and episodic memory function in the prevention and recovery from PTSD, have led Miller et al. (2019) to develop a trauma management and personal resilience training program for police officers that has demonstrated notable effectiveness [181]. The enhancement of hippocampal functioning in the prevention and recovery from PTSD is another area warranting further investigation.

Future studies exploring the modification of subcortical brain activity secondary to implicit automatic nonconscious learning strategies involving repeated exposure to safety-based and secure attachment imagery are needed to confirm the impressions offered here. Moreover, delineating how brain network modifications via neurofeedback and its potential interactions with implicit learning methods will likely offer further guidance for the practice of neuroscience-informed methods of psychotherapy.

Overall, optimal psychotherapy and memory oriented approaches to the treatment of PTSD and dissociation will likely include nonconscious learning methods along with conscious intended processes.

**Funding:** This research received no external funding.

**Institutional Review Board Statement:** Not applicable.

**Informed Consent Statement:** Not applicable.

**Acknowledgments:** I would like to express my appreciation to my daughter, Lindsey M. Damis, for her emotional and instrumental support during the writing of this article. I would also like to acknowledge the professional encouragement, wisdom, and assistance in gathering of articles for this manuscript from my respected colleague, Akira Otani. Finally, I would like to thank Jessica K. Miller for her thoughtful review of an earlier version of this manuscript.

**Conflicts of Interest:** The author declares no conflict of interest.

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
