# Peer review of "The Role of Implicit Memory in the Development and Recovery from Trauma-Related Disorders"

_neurosci, doi:10.3390/neurosci3010005_

Round 1
Reviewer 1 Report
Dr. Damis, did a comprehensive and masterful review of the literature and provided evidence for his thesis.
Author Response
Thank you for your feedback.
Dr. Damis
Reviewer 2 Report
Thank you for this very welcome contribution to understanding of the role of implicit memory and its implications for trauma recovery. This is particularly pertinent given the widespread concern of the pressure on mental health resources to deal with the trauma-related impact of Covid 19.
There are a handful of points on clarity but a few more pointing to other research on the nature of implicit memory and trauma recovery - which will be presented here chronologically.
L37 There is a study (sorry, its my own!) that revealed an associative learning bias as a result of PTSD and impairment in declarative (hippocampal dependent) processing after trauma exposure without PTSD. If this is useful to you it can be found here (the work was co-supervised by Prof Chris Brewin): Impairment in active navigation from trauma and Post-Traumatic Stress Disorder - ScienceDirect
L44 The spatial component of disorganised recall is addressed in a fascinating case study with veterans with PTSD by Kaur et al. which can be found here: Kaur2016.pdf (kcl.ac.uk)
L136 There are studies which show that those with 'trauma histories' and ongoing exposure (namely, police officers) also demonstrate different activation patterns within the vmPFC and limbic system which might be worth a look at, namely one by Peres: Police officers under attack: Resilience implications of an fMRI study - ScienceDirect
L139 Does the 1985 description of implicit memory demonstrate its longstanding role in neuroscience or is there a more modern reference which is more pertinent to your review?
L145 Perhaps one could note (even in a footnote) the significance of activity-dependent hippocampal plasticity and its role in trauma recovery? There are lots of studies out there which I have reviewed in 2016 but there may be more since, see the bibliography here: The Impact of the Brain-Derived Neurotrophic Factor Gene on Trauma and Spatial Processing - PubMed (nih.gov)
L154-276 Might it be useful to have either: a) a more clear explanation for the reader that these themes will be described in full before relating them to trauma or b) more references to why each theme is relevant for trauma within each section or c) a diagram showing the connection between the themes and trauma processing? It does leave the reader chomping at the bit to understand why these are so relevant and does leave the reader somewhat alone, trying to piece it all together.
L216 There is some interesting research on why the motor system is so relevant for hippocampal processing which may be of use here. From the 2010s many hippocampal experiments were undertaken using a treadmill to maximise activity, including studies on trauma. It might be worth looking into.
L273 I think more could be said about fear triggering and trauma here, such as startle response. There is so much research into this I think it's important to acknowledge and consider its implication for this enquiry.
L277 Does it need to be spelt out to the reader that EEG's don't have the capacity to reveal activity in deeper limbic structures or does one assume the audience would know this?
L348 Trauma survivors have also been shown to have better declarative memory in terms of spatial processing (they are more accurate at describing their ability to navigate using the hippocampus than those who have not had any experiences of trauma, suggesting a positive effect). I have not managed to publish this data yet but I do refer to it in this forthcoming publication, which I can send you: Policy Press | The Policing Mind - Developing Trauma Resilience for a New Era, By Jessica K. Miller (bristoluniversitypress.co.uk)
L426 There are some interesting studies on ACEs ALS and the DMN, one of which is here: Early life stress is associated with greater default network deactivation during working memory in healthy controls: a preliminary report | SpringerLink
L450 We found the same, here: Impairment in active navigation from trauma and Post-Traumatic Stress Disorder - ScienceDirect
.... the implications of this feature here: Predictors of PTSD and CPTSD in UK firefighters (tandfonline.com) ie Without timely hippocampal activation after trauma exposure, the ability to process therefore is impaired, leading to higher levels of PTSD in occupations such as Emergency Response. See also the intervention to address this here: Can police be trained in trauma processing to minimise PTSD symptoms? Feasibility and proof of concept with a newly recruited UK police population - Jessica K Miller, Alexandra Peart, Magdalena Soffia, 2020 (sagepub.com)
L444 It's not particularly clear why somatic memory is being discussed with reference to dissociation? Perhaps more could be said about the impact of dissociation on the individual's ability to regulate their physical stress response after prolonged trauma?
L492 I think it needs to be more overtly stated that there needs to be an integration of implicit and declarative memory through contextualisation.
L618 Does there need to be a brief reference to brain development (ie time taken for hippocampus to develop to be able to provide declarative function?)
L707 A visual guide to these neural mechanisms would work really well here.
L726 I'm aware of many studies pertaining to face perception, expression perception and fear in relation to the amygdala and vagus nerve simulation which might work well here. Again, will be in: Policy Press | The Policing Mind - Developing Trauma Resilience for a New Era, By Jessica K. Miller (bristoluniversitypress.co.uk)
L754 Great neuroscience studies on compassion meditation and hippocampal density and Pfc changes in texts such as these:
Mindfulness practice leads to increases in regional brain gray matter density (nih.gov)
Hanson, R. (2020) Neurodharma,
Davidson, R.J., Kabat-Zinn, J., Schumacher, J., Rosenkranz, M., Muller, D., Santorelli, S.F., Urbanowski, F., Harrington, A. Bonus, K., Sheridan, J.F. (2003). Alterations in Brain and Immune Function Produced by Mindfulness Meditation. Psychosomatic Medicine 65, 564 –570.
L888 More on the ability to train the hippocampus into better functionality might be good eg Eleanour Maguire's cab driver study: London taxi drivers and bus drivers: a structural MRI and neuropsychological analysis - PubMed (nih.gov)
All in all, a great review. Apologies for references to my own work but we have a lot in common.
Kind regards.
Author Response
Dr. Miller,
Thank you for your thoughtful and helpful review of my paper. I tried emailing you regarding the article you offered but the address I found failed. I asked the editor for you email noting that I wanted to get the Police article from you but they did not provide it and told me to finish my revisions without it.
Please note my responses to your comments below.
Thank you for this very welcome contribution to understanding of the role of implicit memory and its implications for trauma recovery. This is particularly pertinent given the widespread concern of the pressure on mental health resources to deal with the trauma-related impact of Covid 19.
There are a handful of points on clarity but a few more pointing to other research on the nature of implicit memory and trauma recovery - which will be presented here chronologically.
L37 There is a study (sorry, its my own!) that revealed an associative learning bias as a result of PTSD and impairment in declarative (hippocampal dependent) processing after trauma exposure without PTSD. If this is useful to you it can be found here (the work was co-supervised by Prof Chris Brewin): Impairment in active navigation from trauma and Post-Traumatic Stress Disorder – ScienceDirect
I enjoyed reading your article and added it here as recommended.
L44 The spatial component of disorganised recall is addressed in a fascinating case study with veterans with PTSD by Kaur et al. which can be found here: Kaur2016.pdf (kcl.ac.uk)
This was interesting.
L136 There are studies which show that those with 'trauma histories' and ongoing exposure (namely, police officers) also demonstrate different activation patterns within the vmPFC and limbic system which might be worth a look at, namely one by Peres: Police officers under attack: Resilience implications of an fMRI study – ScienceDirect
I wanted to keep the focus of the article on the role of implicit memory and addressed the role of the vmPFC in other sections of the paper.
L139 Does the 1985 description of implicit memory demonstrate its longstanding role in neuroscience or is there a more modern reference which is more pertinent to your review?
So much of the work on implicit memory was in the late 80s and throughout the 90s. Graf & Schacter (1985) was such a seminal work that I wanted to start with it. There are subsequent references that were more recent.
L145 Perhaps one could note (even in a footnote) the significance of activity-dependent hippocampal plasticity and its role in trauma recovery? There are lots of studies out there which I have reviewed in 2016 but there may be more since, see the bibliography here: The Impact of the Brain-Derived Neurotrophic Factor Gene on Trauma and Spatial Processing - PubMed (nih.gov)
I addressed this in the discussion
L154-276 Might it be useful to have either: a) a more clear explanation for the reader that these themes will be described in full before relating them to trauma or b) more references to why each theme is relevant for trauma within each section or c) a diagram showing the connection between the themes and trauma processing? It does leave the reader chomping at the bit to understand why these are so relevant and does leave the reader somewhat alone, trying to piece it all together.
All good recommendations – I added a comment at the beginning explaining the nature of the overview.
L216 There is some interesting research on why the motor system is so relevant for hippocampal processing which may be of use here. From the 2010s many hippocampal experiments were undertaken using a treadmill to maximise activity, including studies on trauma. It might be worth looking into.
What I found was related to treadmill activity and prevention of decreased hippocampal functioning. I was not able to find the topic you recommended.
L273 I think more could be said about fear triggering and trauma here, such as startle response. There is so much research into this I think it's important to acknowledge and consider its implication for this enquiry.
I choose not to elaborate more on fear conditioning as I covered it to some extent at various points in the paper and I wanted to keep the focus on implicit memory.
L277 Does it need to be spelt out to the reader that EEG's don't have the capacity to reveal activity in deeper limbic structures or does one assume the audience would know this?
Comment added.
L348 Trauma survivors have also been shown to have better declarative memory in terms of spatial processing (they are more accurate at describing their ability to navigate using the hippocampus than those who have not had any experiences of trauma, suggesting a positive effect). I have not managed to publish this data yet but I do refer to it in this forthcoming publication, which I can send you: Policy Press | The Policing Mind - Developing Trauma Resilience for a New Era, By Jessica K. Miller (bristoluniversitypress.co.uk)
I look forward to seeing those data.
L426 There are some interesting studies on ACEs ALS and the DMN, one of which is here: Early life stress is associated with greater default network deactivation during working memory in healthy controls: a preliminary report | SpringerLink
There is a lot of interesting work on disruption in the DMN but I chose to limit the focus of this paper to subcortical matters and not get into brain networks other to mention it in the discussion.
L450 We found the same, here: Impairment in active navigation from trauma and Post-Traumatic Stress Disorder - ScienceDirect
.... the implications of this feature here: Predictors of PTSD and CPTSD in UK firefighters (tandfonline.com) ie Without timely hippocampal activation after trauma exposure, the ability to process therefore is impaired, leading to higher levels of PTSD in occupations such as Emergency Response. See also the intervention to address this here: Can police be trained in trauma processing to minimise PTSD symptoms? Feasibility and proof of concept with a newly recruited UK police population - Jessica K Miller, Alexandra Peart, Magdalena Soffia, 2020 (sagepub.com)
Not able to access more than the abstract but mentioned it in the discussion.
L444 It's not particularly clear why somatic memory is being discussed with reference to dissociation? Perhaps more could be said about the impact of dissociation on the individual's ability to regulate their physical stress response after prolonged trauma?
This is a quote from van der Kolk, he is really referring to sensory memory.
L492 I think it needs to be more overtly stated that there needs to be an integration of implicit and declarative memory through contextualisation.
L618 Does there need to be a brief reference to brain development (ie time taken for hippocampus to develop to be able to provide declarative function?)
References added.
L707 A visual guide to these neural mechanisms would work really well here.
L726 I'm aware of many studies pertaining to face perception, expression perception and fear in relation to the amygdala and vagus nerve simulation which might work well here. Again, will be in: Policy Press | The Policing Mind - Developing Trauma Resilience for a New Era, By Jessica K. Miller (bristoluniversitypress.co.uk)
Not able to access your article.
L754 Great neuroscience studies on compassion meditation and hippocampal density and Pfc changes in texts such as these:
These are great topics and worth papers in themselves. I don’t know if I could speculate on the relationship of these changes to the primary point of the paper regarding the safety and ideal parent figure imagery techniques to modify implicit memory per se.
Mindfulness practice leads to increases in regional brain gray matter density (nih.gov)
Hanson, R. (2020) Neurodharma,
I purchased this and look forward to reading it!
Davidson, R.J., Kabat-Zinn, J., Schumacher, J., Rosenkranz, M., Muller, D., Santorelli, S.F., Urbanowski, F., Harrington, A. Bonus, K., Sheridan, J.F. (2003). Alterations in Brain and Immune Function Produced by Mindfulness Meditation. Psychosomatic Medicine 65, 564 –570.
L888 More on the ability to train the hippocampus into better functionality might be good eg Eleanour Maguire's cab driver study: London taxi drivers and bus drivers: a structural MRI and neuropsychological analysis - PubMed (nih.gov)
All in all, a great review. Apologies for references to my own work but we have a lot in common.
Kind regards.
Thank you for your extensive review and helpful suggestions. If I didn’t develop some of your suggestions it was because I wanted to stay on track with the conclusion I was moving toward. However, I see that I could have done more with the contextualization/integration part.
All my best,
Louis Damis (drdamis@louisdamisphd.com)
Reviewer 3 Report
The review titled " The Role of Implicit Memory in the Development and Recovery from Trauma-Related Disorders " by Damis et al, evaluate that the modification of implicit memory can promote the recovery from post-traumatic related disorders thus highlighting the importance of implicit memory over explicit memory. The overall review was well explained and structured. I believe this review will be of interest to the readership of Trauma-Related Disorders since the authors summarized key past developments and provide their view on future perspectives for the field. However, a minor correction needs to be considered, the paragraph under 2.3 and 3.0 are not well aligned.
Author Response
Thank you for your feedback. I was unable to correct some document layout features but hopefully, the editor will be able to fix them.
Dr. Damis